# UNDISTILLABLE: MAKING A NASTY TEACHER THAT CANNOT TEACH STUDENTS

**Haoyu Ma[1], Tianlong Chen[2], Ting-Kuei Hu[3], Chenyu You[4], Xiaohui Xie[1], Zhangyang Wang[2]**
[1]University of California, Irvine, [2]University of Texas at Austin,
[3]Texas A&M University, [4]Yale University
`{haoyum3,xhx}@uci.edu,{tianlong.chen, atlaswang}@utexas.edu,`
`tkhu@tamu.edu, chenyu.you@yale.edu`

## ABSTRACT

Knowledge Distillation (KD) is a widely used technique to transfer knowledge from pre-trained teacher models to (usually more lightweight) student models. However, in certain situations, this technique is more of a curse than a blessing. For instance, KD poses a potential risk of exposing intellectual properties (IPs): even if a trained machine learning model is released in "black boxes" (e.g., as executable software or APIs without open-sourcing code), it can still be replicated by KD through imitating input-output behaviors. To prevent this unwanted effect of KD, this paper introduces and investigates a concept called *Nasty Teacher*: a specially trained teacher network that yields nearly the same performance as a normal one, but would significantly degrade the performance of student models learned by imitating it. We propose a simple yet effective algorithm to build the nasty teacher, called *self-undermining knowledge distillation*. Specifically, we aim to maximize the difference between the output of the nasty teacher and a normal pre-trained network. Extensive experiments on several datasets demonstrate that our method is effective on both standard KD and data-free KD, providing the desirable KD-immunity to model owners for the first time. We hope our preliminary study can draw more awareness and interest in this new practical problem of both social and legal importance. Our codes and pre-trained models can be found at `https://github.com/VITA-Group/Nasty-Teacher`.

## 1 INTRODUCTION

Knowledge Distillation (KD) (Hinton et al., 2015) aims to transfer useful knowledge from a teacher neural network to a student network by imitating the input-output behaviors. The student model imitates logit outputs or activation maps from the teacher by optimizing a distillation loss. The efficacy of leveraging teacher knowledge to boost student performance has been justified in many application fields (Wang et al., 2019; Liu et al., 2019; Chen et al., 2020b;a), yielding high performance and often lighter-weight student models. Typically KD requires learning the student model over the same training set used to train the teacher. However, recent studies (Lopes et al., 2017; Chen et al., 2019) have demonstrated the feasibility of the data-free knowledge distillation, in which the knowledge is transferred from the teacher to the student without accessing the same training data. This is possible because the training data are implicitly encoded in the trained weights of deep neural nets. The data-free knowledge distillation is able to inversely decode and re-synthesize the training data from the weights, and then clone the input-output mapping of the teacher network.

With many practical benefits brought by the KD technique, this paper looks at KD's unwanted severe side effect, that it might pose risks to the machine learning intellectual property (IP) protection. Many machine learning models will be only released as executable software or APIs, without open-sourcing model configuration files or codes, e.g., as "black boxes". That can be due to multifold reasons, such as (1) those advanced models might take huge efforts and resources for the model owners to develop, who would need to keep this technical barrier; or (2) those trained models have involved protected training data or other information, that are legally or ethically prohibited to be openly shared. However, KD techniques might open a loophole to unauthorized infringers to clone the IP model's functionality, by simply imitating the black box's input and output behaviors (leaked

knowledge). The feasibility of data-free KD (Chen et al., 2019; Yin et al., 2020) eliminates the necessity of accessing original training data, therefore making this cloning more practically feasible. Even worse, those techniques point to reverse-engineering ways (Yin et al., 2020) to recover the (potentially private) training data from black-box models, threatening the owners' data privacy and security (Yonetani et al., 2017; Wu et al., 2018).

To alleviate the issue, this paper introduces a defensive approach for model owners, called *Nasty Teacher*. A nasty teacher is a specially trained network that yields nearly the same performance as a normal one; but if used as a teacher model, it will significantly degrade the performance of student models that try to imitate it. In general, the concept of nasty teacher is related to the **backdoor attack** on deep learning systems (Chen et al., 2017), which creates a model to fit "adversarial" goals in an "imperceptible" way. However, while backdoor attacks aim to manipulate or damage the performance of the poisoned model itself when triggered by specific inputs, the goal of the nasty teacher is to undermine the performance of any student network derived from it. The primary objective of constructing a nasty teacher is for model protection - a novel motivation and setting that have not been explored before. Our contributions are summarized as follows:

- We introduce the novel concept of *Nasty Teacher*, a defensive approach to prevent knowledge leaking and unauthorized model cloning through KD without sacrificing performance. We consider it a promising first step towards machine learning IP and privacy protection.

- We propose a simple yet efficient algorithm, called *self-undermining knowledge distillation*, to directly build a nasty teacher through self-training, requiring no additional dataset nor auxiliary network. Specifically, the model is optimized by maximizing the difference between the nasty teacher (the desired one) and a normally trained counterpart.

- We conduct extensive experiments on both standard KD and data-free KD approaches, and demonstrate that nasty teacher trained by self-undermining KD can achieve nearly the same accuracy as their original counterpart (less than 1% accuracy gap), while the student model learned from it will degrade accuracy by up to over 10% or even diverge during training.

## 2 RELATED WORK

### 2.1 KNOWLEDGE DISTILLATION

Knowledge distillation aims to boost the performance of light-weight models (students) under the guidance of well-trained complicated networks (teachers). It is firstly introduced in (Hinton et al., 2015), where the student directly mimics the soft probabilities output produced by the well pretrained teacher. The following researchers explore the knowledge transferal from either intermediate features (Romero et al., 2014; Zagoruyko & Komodakis, 2016; Passalis & Tefas, 2018; Ahn et al., 2019; Li et al., 2020), or logit responses (Park et al., 2019; Mirzadeh et al., 2019; Chen et al., 2021a;b; Ma et al., 2021). Recent studies have also shown that, instead of distilling from a complicated teacher, the student networks can even be boosted by learning from its own pre-trained version (Furlanello et al., 2018; Zhang et al., 2019; Yun et al., 2020; Yuan et al., 2020).

Several recent works also focus on data-free knowledge distillation, under which settings students are not able to access the data used to train teachers. In (Lopes et al., 2017), the author attempts to reconstruct input data by exploring encoded meta-data lying in the pre-trained teacher network. In the following work, the author of (Chen et al., 2019) proposes a learning scheme, called "Data-Free Learning" (DAFL), which treats the teacher as a fixed discriminator, and jointly trains a generator to synthesize training examples so that maximum responses could be obtained on the discriminator. The latest work "DeepInversion" (Yin et al., 2020) directly synthesizes input images given random noise by "inverting" a trained network. Specifically, their method optimizes the input random noise into high-fidelity images with a fixed pre-trained network (teacher).

### 2.2 POISONING ATTACK ON NEURAL NETWORK

The typical goal of poisoning attack is to degrade the accuracy of models by injecting poisoned data into training set (Xiao et al., 2015; Moosavi-Dezfooli et al., 2016). On the contrary, backdoor attack intends to open a loophole (usually unperceived) to the model via inserting well-crafted malicious data into training set (Chen et al., 2017; Gu et al., 2017; Kurita et al., 2020). The goal of back-

door attack is to make the poisoned model perform normally well for most of the time, yet failing specifically when attacker-triggered signals are given.

Our proposed self-undermining knowledge distillation aims to create a special teacher model (i.e., an undistillable model), which normally performs by itself but "triggers to fail" only when being mimiced through KD. The motivation looks similar to backdoor attack's at the first glance, but differs in the following aspects. Firstly, the backdoor attack can only be triggered by pre-defined patterns, while our nasty teachers target at degrading any arbitrary student network through KD. Secondly, backdoor attack tends to poison the model itself, while our nasty teacher aims to undermine other student networks while preservingits own performance. Thirdly, our goal is to prevent knowledge leaking in order to protect released IP, as a defensive point of view, while the backdoor attack tends to break down the system by triggering attacking signals, as an attacking point of view.

### 2.3 PROTECTION OF MODEL IP

Due to the commercial value, IP protection for deep networks has drawn increasing interests from both academia and industry. Previous methods usually rely on watermark-based (Uchida et al., 2017; Zhang et al., 2020a) or passport-based (Fan et al., 2019; Zhang et al., 2020b) ownership verification methods to protect the IP. Nevertheless, these methods can only detect IP infringement but remain ineffective to avoid model cloning.

A few recent works also explore defensive methods against model stealing (Kariyappa & Qureshi, 2020; Juuti et al., 2019; Orekondy et al., 2020). Typically, they assume attackers obtain pseudo-labels on their own synthetic data or surrogate data by querying the black-box model, and train a network on the new dataset to clone the model. However, none of these defense approaches have explored the KD-based model stealing, which is rather a practical threat.

## 3 METHODOLOGY

### 3.1 REVISITING KNOWLEDGE DISTILLATION

Knowledge distillation (Hinton et al., 2015) helps the training process of "student" networks by distilling knowledge from one or multiple well-trained "teacher" networks. The key idea is to leverage soft probabilities output of teacher networks, of which incorrect-class assignments reveal the way how teacher networks generalize from previous training. By mimicking probabilities output, student networks are able to imbibe the knowledge that teacher networks have discovered before, and the performance of student networks is usually better than those being trained with labels only. In what follows, we formally formulate the learning process of knowledge distillation.

Given a pre-trained teacher network $f_{\theta_T}(\cdot)$ and a student network $f_{\theta_S}(\cdot)$, where $\theta_T$ and $\theta_S$ denote the network parameters, the goal of knowledge distillation is to force the output probabilities of $f_{\theta_S}(\cdot)$ to be close to that of $f_{\theta_T}(\cdot)$. Let $(x_i, y_i)$ denote a training sample in dataset $\mathcal{X}$ and $p_{f_\theta}(x_i)$ indicate the logit response of $x_i$ from $f_\theta(\cdot)$, the student network $f_{\theta_S}$ could be learned by the following:

$$\min_{\theta_S} \sum_{(x_i, y_i) \in \mathcal{X}} \alpha \tau_s^2 \mathcal{KL}(\sigma_{\tau_s}(p_{f_{\theta_T}}(x_i)), \sigma_{\tau_s}(p_{f_{\theta_S}}(x_i))) + (1 - \alpha)\mathcal{XE}(\sigma(p_{f_{\theta_S}}(x_i)), y_i), \quad (1)$$

where $\mathcal{KL}(\cdot, \cdot)$ and $\mathcal{XE}(\cdot, \cdot)$ are Kullback-Leibler divergence (K-L divergence) and cross-entropy loss, respectively. The introduced "softmax temperature" function $\sigma_{\tau_s}(\cdot)$ (Hinton et al., 2015) produces soft probabilities output when a large temperature $\tau_s$ (usually greater than 1) is picked, and it decays to normal softmax function $\sigma(\cdot)$ when $\tau_s$ equals 1. Another hyper-parameter $\alpha$ is also introduced to balance between knowledge distillation and cost minimization.

### 3.2 TRAINING NASTY TEACHERS: RATIONALE AND IMPLEMENTATION

**Rationale** The goal of nasty teacher training endeavors to create a special teacher network, of which performance is nearly the same as its normal counterpart, that any arbitrary student networks *cannot* distill knowledge from it. To this end, we propose a simple yet effective algorithm, dubbed *Self-Undermining Knowledge Distillation*, while maintaining its correct class assignments, maximally disturbing its in-correct class assignments so that no beneficial information could be distilled from it, as described next.

Let $f_{\theta_T}(\cdot)$ and $f_{\theta_A}(\cdot)$ denote the desired nasty teacher and its adversarial learning counterpart, the self-undermining training aims to maximize the K-L divergence between the adversarial network and the nasty teacher one, so that a false sense of generalization could be output from the nasty teacher. The learning process of the nasty teacher could be formulated as follows,

$$\min_{\theta_T} \sum_{(x_i, y_i) \in \mathcal{X}} \mathcal{XE}(\sigma(p_{f_{\theta_T}}(x_i)), y_i) - \omega \tau_A^2 \mathcal{KL}(\sigma_{\tau_A}(p_{f_{\theta_T}}(x_i)), \sigma_{\tau_A}(p_{f_{\theta_A}}(x_i))), \qquad (2)$$

where the former term aims to maintain the accuracy of the nasty teacher by minimizing the cross entropy loss, and the latter term achieves the "undistillability" by maximizing KL divergence between the nasty teacher and the adversarial one. Similarly in equation 1, $\tau_A$ denotes the temperature for self-undermining, and $\omega$ balances the behavior between normal training and adversarial learning.

**Implementation** We naturally choose the same network architecture for $f_{\theta_T}(\cdot)$ and $f_{\theta_A}(\cdot)$ (yet different group of network parameters) since no additional assumption on network architecture is made here. We provide a throughout study with respect to the selection of architectures in 4.3, revealing how the architecture of $f_{\theta_A}(\cdot)$ influences the adversarial training. As for the update rule, the parameter of $f_{\theta_A}(\cdot)$ is typically normally **pre-trained in advance** and **fixed** during adversarial training, and only the parameter of $f_{\theta_T}(\cdot)$ is updated. Note that the selected temperature $\tau_A$ does not to be necessary the same as $\tau_s$ in equation 1, and we provide a comprehensive study with respect to $\tau_s$ in 4.3. Once upon finishing adversarial training, the nasty teacher $f_{\theta_T}(\cdot)$ could be released within defense of KD-based model stealing.

# 4 EXPERIMENTS

## 4.1 NASTY TEACHER ON STANDARD KNOWLEDGE DISTILLATION

To evaluate the effectiveness of our nasty teachers, we firstly execute self-undermining training to create nasty teachers based on equation 2. Then, given an arbitrary student network, we evaluate the performance of nasty teachers by carrying out knowledge distillation from equation 1 to recognize how much nasty teachers go against KD-based model stealing.

### 4.1.1 EXPERIMENTAL SETUP

**Network.** We explore the effectiveness of our nasty teachers on three representative datasets, i.e., CIFAR-10, CIFAR-100, and Tiny-ImageNet. Firstly, we consider ResNet-18 (teacher network) and 5-layer plain CNN (student network) as our baseline experiment in CIFAR-10, and replace the student network with two simplified ResNets designed for CIFAR-10 (He et al., 2016), i.e., ResNetC-20 and ResNetC-32, to explore the degree of impact with respect to the capacity of student networks. For both CIFAR-100 and Tiny-ImageNet, we follow the similar setting in (Yuan et al., 2020), where three networks from ResNet family, i.e., ResNet-18, ResNet-50 and ResNeXt-29, are considered as teacher networks, and three widely used light-weight networks, i.e., MobileNetV2 (Sandler et al., 2018), ShuffleNetV2 (Ma et al., 2018) and ResNet-18, are served as student networks. Following the "self-KD" setting in (Yuan et al., 2020), an additional comparative experiment, dubbed "Teacher Self", is provided, where the architectures of the student and the teacher are set to be identical.

**Training.** The distilling temperature $\tau_A$ for self-undermining training is set to 4 for CIFAR-10 and 20 for both CIFAR-100 and Tiny-ImageNet as suggested in (Yuan et al., 2020). For the selection of $\omega$, 0.004, 0.005, and 0.01 are picked for CIFAR-10, CIFAR-100, and Tiny-ImageNet, respectively. For the plain CNN, we train it with a learning rate of $1e-3$ for 100 epochs and optimize it by Adam optimizer (Kingma & Ba, 2014). Other networks are optimized by SGD optimizer with momentum 0.9 and weight decay $5e-4$. The learning rate is initialized as 0.1. Networks are trained by 160 epochs with learning rate decayed by a factor of 10 at the 80th and 120th epoch for CIFAR-10, and 200 epochs with learning rate decayed by a factor of 5 at the 60th, 120th and 160th epoch for CIFAR-100 and Tiny-ImageNet. Without specifically mentioned, the temperature $\tau_s$ is the same as $\tau_A$, which is used for self-undermining training.

### 4.1.2 EXPERIMENTAL RESULTS

Our experimental results on CIFAR-10, CIFAR-100, and Tiny-ImageNet are presented in Table 1, Table 2 and Table 3, respectively. Firstly, we observe that all nasty teachers still perform similarly

as their normal counterparts by at most $\sim 2\%$ accuracy drop. Secondly, in contrast to normally trained teacher networks, from which the accuracy of student networks could be boosted by at most $\sim 4\%$ by distilling, no student network can benefit from distilling the knowledge of nasty teachers. It indicates that our nasty teachers could successfully provide a false sense of generalization to student networks, resulting in decreases of accuracy by $1.72\%$ to $67.57\%$. We also notice that weak student networks (e.g. MobilenetV2) could be much more poisoned from distilling toxic knowledge than stronger networks (e.g., ResNet-18), since light-weight networks intend to rely more on the guidance from teachers. It terms out that KD-based model stealing is no longer practical if the released model is "nasty" in this sense. Additional experiments, dubbed "Teacher Self", are also provided here. Opposite to the conclusion drew in (Yuan et al., 2020), the student network still cannot be profited from the teacher network even if their architectures are exactly the same. The aforementioned experiments have justified the efficacy of the proposed self-undermining training.

Table 1: Experimental results on CIFAR-10.

| Teacher network | Teacher performance | Students performance after KD | | | |
|---|---|---|---|---|---|
| | | CNN | ResNetC-20 | ResNetC-32 | ResNet-18 |
| Student baseline | - | 86.64 | 92.28 | 93.04 | 95.13 |
| ResNet-18 (normal) | 95.13 | 87.75 (+1.11) | 92.49 (+0.21) | 93.31 (+0.27) | 95.39 (+0.26) |
| ResNet-18 (nasty) | 94.56 (-0.57) | 82.46 (-4.18) | 88.01 (-4.27) | 89.69 (-3.35) | 93.41 (-1.72) |

Table 2: Experimental results on CIFAR-100.

| Teacher network | Teacher performance | Students performance after KD | | | |
|---|---|---|---|---|---|
| | | Shufflenetv2 | MobilenetV2 | ResNet-18 | Teacher Self |
| Student baseline | - | 71.17 | 69.12 | 77.44 | - |
| ResNet-18 (normal) | 77.44 | 74.24 (+3.07) | 73.11 (+3.99) | 79.03 (+1.59) | 79.03 (+1.59) |
| ResNet-18 (nasty) | 77.42(-0.02) | 64.49 (-6.68) | 3.45 (-65.67) | 74.81 (-2.63) | 74.81 (-2.63) |
| ResNet-50 (normal) | 78.12 | 74.00 (+2.83) | 72.81 (+3.69) | 79.65 (+2.21) | 80.02 (+1.96) |
| ResNet-50 (nasty) | 77.14 (-0.98) | 63.16 (-8.01) | 3.36 (-65.76) | 71.94 (-5.50) | 75.03 (-3.09) |
| ResNeXt-29 (normal) | 81.85 | 74.50 (+3.33) | 72.43 (+3.31) | 80.84 (+3.40) | 83.53 (+1.68) |
| ResNeXt-29 (nasty) | 80.26(-1.59) | 58.99 (-12.18) | 1.55 (-67.57) | 68.52 (-8.92) | 75.08 (-6.77) |

Table 3: Experimental results on Tiny-ImageNet

| Teacher network | Teacher performance | Students performance after KD | | | |
|---|---|---|---|---|---|
| | | Shufflenetv2 | MobilenetV2 | ResNet-18 | Teacher Self |
| Student baseline | - | 55.74 | 51.72 | 58.73 | - |
| ResNet-18 (normal) | 58.73 | 58.09 (+2.35) | 55.99 (+4.27) | 61.45 (+2.72) | 61.45 (+2.72) |
| ResNet-18 (nasty) | 57.77 (-0.96) | 23.16 (-32.58) | 1.82 (-49.90) | 44.73 (-14.00) | 44.73 (-14.00) |
| ResNet-50 (normal) | 62.01 | 58.01 (+2.27) | 54.18 (+2.46) | 62.01 (+3.28) | 63.91 (+1.90) |
| ResNet-50 (nasty) | 60.06 (-1.95) | 41.84 (-13.90) | 1.41 (-50.31) | 48.24 (-10.49) | 51.27 (-10.74) |
| ResNeXt-29 (normal) | 62.81 | 57.87 (+2.13) | 54.34 (+2.62) | 62.38 (+3.65) | 64.22 (+1.41) |
| ResNeXt29 (nasty) | 60.21 (-2.60) | 42.73 (-13.01) | 1.09 (-50.63) | 54.53 (-4.20) | 59.54 (-3.27) |

## 4.2 QUALITATIVE ANALYSIS

We present visualizations of our nasty ResNet-18 and the normal ResNet-18 on CIFAR-10 to qualitatively analyze the behavior of nasty teachers. Figure 1 visualizes the logit responses of normal ResNet-18 and its nasty counterpart after "softmax temperature" function. We notice that the logit response of nasty ResNet-18 consists of multiple peaks, where normal ResNet-18 consistently outputs a single peak. We observe that the class-wise multi-peak responses might be un-correlated

to the sense of generalization. For instance, the class-wise output of bird and dog might both respond actively, and it will give a false sense of generalization to student networks. We hypothesize that the multi-peak logits misleads the learning from the knowledge distillation and degrades the performance of students.

We also present the visualizations of t-Distributed Stochastic Neighbor Embedding (t-SNE) for both feature embeddings and output logits, as illustrated in Figure 2. It is observed that the feature-space inter-class distance of nasty ResNet-18 and the normal ResNet-18 behaves similarly, which aligns with our goal that nasty teachers should perform similarly to their normal counterparts. Meanwhile, we also observe that the logit response of nasty ResNet-18 has been heavily shifted, and it entails that our method mainly modifies the weights in the final fully-connected layer.

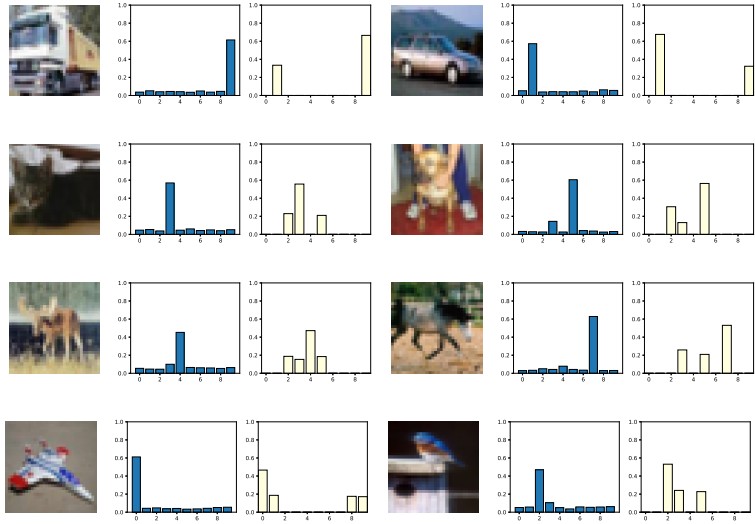

Figure 1: The visualization of logit responses after "temperature softmax" function. Each row represents two examples from CIFAR-10. The sampled images are shown in the 1st and the 4th columns. The 2nd and the 5th columns summarize the scaled output from the normal teacher. The 3rd and the 6th columns represent the scaled output from the nasty teacher

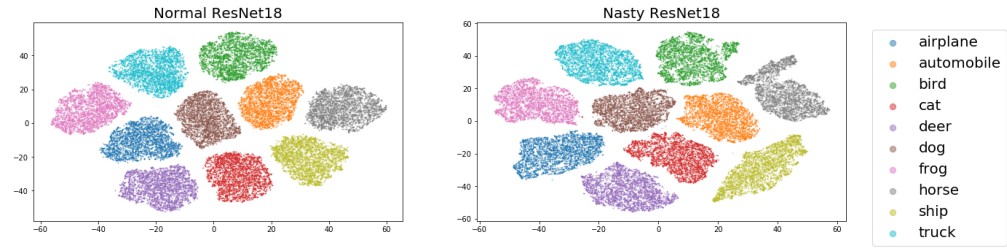

(a) tSNE of feature embeddings before fully-connected layer. The dimension of feature embeddings is $512$.

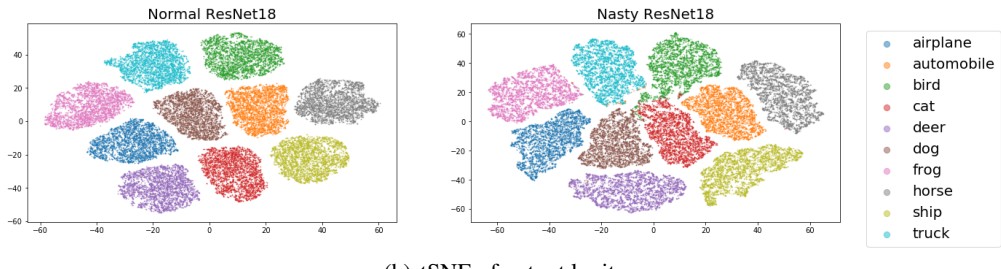

(b) tSNE of output logits

Figure 2: Visualization of tSNEs for both normal and nasty ResNet18 on CIFAR-10. Each dot represents one data point.

## 4.3 ABLATION STUDY

**Adversarial Network.** Instead of choosing the same architecture for both nasty teacher and adversarial network, we vary the architecture of the adversarial network to measure the consequent influence with respect to different network structures. As illustrated in Table 4, our training method shows the generality to various architectures (e.g., ResNet-18, ResNext-29, etc), and we also notice that weak networks (e.g., Plain CNN) might lead to less effective nasty teachers. However, although stronger networks contribute to more effective nasty teachers, we observe that the trade-off accuracy is saturated quickly and converges to "self-undermining" ones. Thus, we consider the "self-undermining" training as a convenient fallback choice.

Table 4: Ablation study w.r.t the architecture of the adversarial network $f_{\theta_A}(\cdot)$ on CIFAR-10.

| Teacher network | Teacher performance | Students after KD | | | |
|---|---|---|---|---|---|
| | | CNN | ResNetC20 | ResNetC32 | ResNet18 |
| Student baseline | - | 86.64 | 92.28 | 93.04 | 95.13 |
| ResNet18(normal) | 95.13 | 87.75 (+1.11) | 92.49 (+0.21) | 93.31 (+0.27) | 95.39 (+0.26) |
| ResNet18(ResNet18) | 94.56 (-0.57) | 82.46 (-4.18) | 88.01 (-4.27) | 89.69 (-3.35) | 93.41 (-1.72) |
| ResNet18(CNN) | 93.82 (-1.31) | 77.12 (-9.52) | 88.32 (-3.96) | 90.40 (-2.64) | 94.05 (-1.08) |
| ResNet18(ResNeXt-29) | 94.55 (-0.58) | 82.75 (-3.89) | 88.17 (-4.11) | 89.48 (-3.56) | 93.75 (-1.38) |

**Students Network.** In practice, the owners of teacher networks have no knowledge of student's architecture, and it is possible that the student is more complicated than the teacher. As the *Reversed KD* in Yuan et al. (2020), the superior network can also be enhanced by learning from a weak network. To explore the generalization ability of our method, we further conduct experiments on the reversed KD. In detail, we consider the ResNet-18 as teacher, and ResNet-50 and ResNeX29 as students. From Table 5, these two sophisticated students can still be slightly improved by distilling from a normal ResNet-18 in most cases, while be degraded by distilling from a nasty ResNet-18. This implies that our method is also effective for the reversed KD setting.

Table 5: Ablation study w.r.t the architecture of the student networks.

| Dataset | CIFAR-10 | | CIFAR-100 | |
|---|---|---|---|---|
| Student network | ResNet-50 | ResNeXt-29 | ResNet-50 | ResNeXt-29 |
| Student baseline | 94.98 | 95.60 | 78.12 | 81.85 |
| KD from ResNet-18 (normal) | 94.45 (-0.53) | 95.92 (+0.32) | 79.94 (+1.82) | 82.14 (+0.29) |
| KD from ResNet-18 (nasty) | 93.13 (-1.85) | 92.20 (-3.40) | 74.28 (-3.84) | 78.88 (-2.97) |

**Weight $\omega$.** As illustrated in Figure 3, we vary the weight $\omega$ from 0 to 0.1 on CIFAR-10 and from 0 to0.01 on CIFAR-100. We show that nasty teachers can degrade the performance of student networks no matter what $\omega$ is selected. By adjusting $\omega$, we could also control the trade-off between performance suffering and nasty behavior. In other words, a more toxic nasty teacher could be learned by picking a larger $\omega$ at the expense of more accuracy loss.

**Temperature $\tau_s$.** By default, the $\tau_s$, used for knowledge distillation, is the same as $\tau_A$, used in the self-undermining training. To explore the impact of temperature selection, we vary the temperature $\tau_s$ from 1 to 20, and Figure 4 presents the accuracy of students after knowledge distillation with the given $\tau_s$. We show that nasty teachers could always degrade the performance of students no matter what $\tau_s$ is picked. Generally, with a larger $\tau_s$, the performance of student networks would be degraded more by the nasty teacher since a larger temperature $\tau_s$ usually lead to more noisy logit outputs. Note that our nasty teacher is still effective even if the student directly distills knowledge from probabilities ($\tau_s$=1).

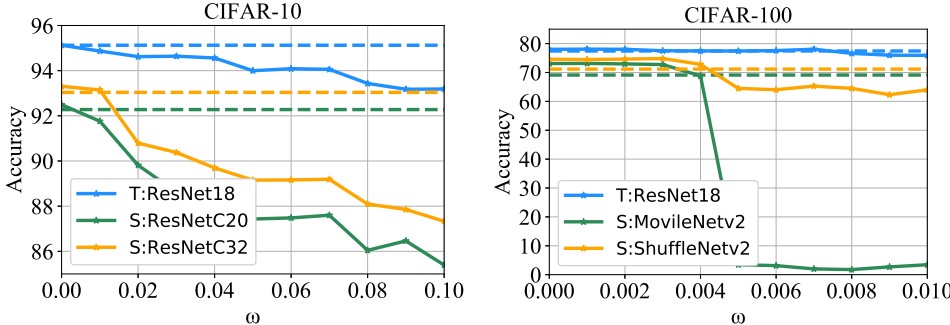

Figure 3: Ablation study w.r.t $\omega$ on CIFAR-10 and CIFAR-100. The initials "T" and "S" in the legend represent teacher networks and student networks, respectively. The dash-line represents the accuracy that the model is normally trained.

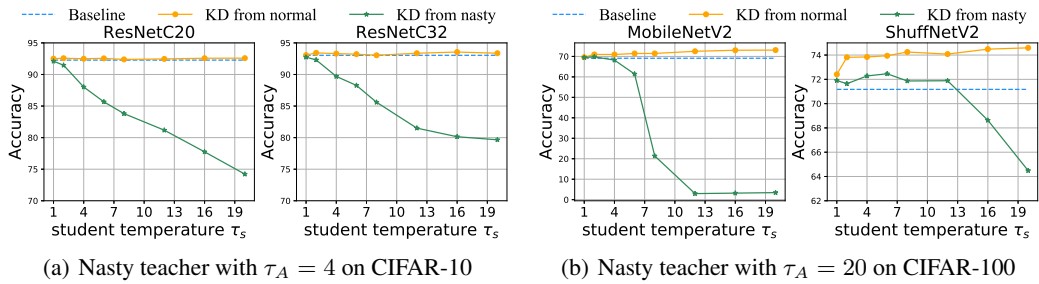

(a) Nasty teacher with $\tau_A = 4$ on CIFAR-10      (b) Nasty teacher with $\tau_A = 20$ on CIFAR-100

Figure 4: Ablation study w.r.t temperature $\tau_s$. The architecture of teacher networks are ResNet-18 for both CIFAR-10 and CIFAR-100 experiments. Each figure presents accuracy curves of student networks under the guidance of the nasty or normal ResNet-18 with various temperature $\tau_s$.

**Balance Factor $\alpha$.** We by default set $\alpha$ to 0.9 as the common practice in (Hinton et al., 2015). To explore the effect of $\alpha$, we conduct the experiments by varying $\alpha$ from 0.1 to 1.0, and the experimental results were summarized in Figure 5(a). In general, our nasty teachers could degrade the performance of student networks regardless of what $\alpha$ is selected. We also observe that a small $\alpha$ can help student networks perform relatively better when distilling from the nasty teacher. However, a small $\alpha$ also makes the student depend less on the teacher's knowledge and therefore benefit less from KD itself. Therefore, the student cannot easily get rid of the nasty defense while still mincing effectively through KD, by simply tuning $\alpha$ smaller.

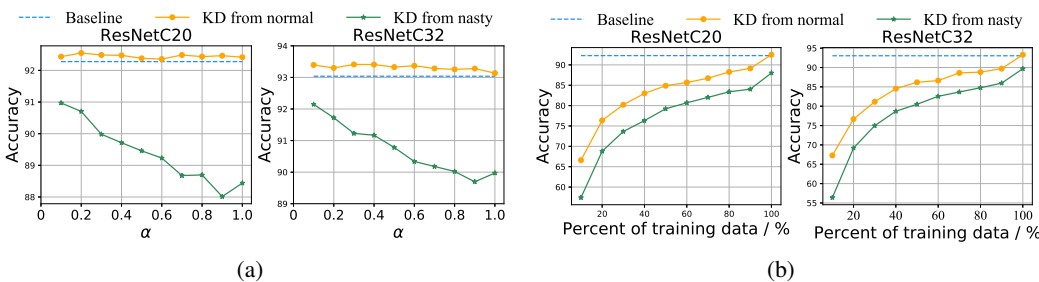

(a)                          (b)

Figure 5: Ablation study with respect to various $\alpha$ (a) and different percentage of training samples (b). Both experiments are conducted under the supervision of either normal or nasty ResNet-18.

**Percentage of Training Samples.** In practice, stealers (student networks) may not have full access to all training examples. Thus, to reflect the practical scenario of model stealing, we conduct the experiments by varying the percentage of training examples from 10% to 90% and keep other hyper-parameters the same. As illustrated in Figure 5(b), comparing to normally trained teachers, the nasty teachers still consistently contribute negatively to student networks.

### 4.4 NASTY TEACHER ON DATA-FREE KNOWLEDGE DISTILLATION

Instead of getting full access to all training samples, KD without accessing any training sample is considered a more realistic way to practice model stealing. To reflect the practical behavior of stealers, we evaluate our nasty teachers on two state-of-the-art data-free knowledge distillation methods, i.e., DAFL (Chen et al., 2019) and DeepInversion (Yin et al., 2020), where students only have access to the probabilities produced by teacher networks. For a fair comparison, we strictly follow the setting in DAFL and adopt ResNet-34 and ResNet-18 as the teacher-student pair for further knowledge distillation. The experiments are conducted on both CIFAR-10 and CIFAR-100, where nasty ResNet-34 is trained by respectively setting $\omega$ and $\tau_A$ as $0.04$ and $4$. The experimental results with regard to DAFL are summarized in Table 6. We show that the nasty ResNet-34 largely detriments the accuracy of student networks by more than $5\%$, in contrast to that under the supervision of normal ResNet-34. Based on DeepInversion, we also present the visualizations in Figure 6, where the images are generated by reverse engineering both nasty and normal ResNet-34. We demonstrate that images generated from normal ResNet-34 enable high visual fidelity, while images from nasty ResNet-34 consist of distorted noises and even false category-wise features. This visualization showcases how nasty teachers prevent illegal data reconstruction from reverse engineering.

Table 6: Data-free KD from nasty teacher on CIFAR-10 and CIFAR-100

| dataset | CIFAR-10 | | CIFAR-100 | |
|---|---|---|---|---|
| Teacher Network | Teacher Accuracy | DAFL | Teacher Accuracy | DAFL |
| ResNet34 (normal) | 95.42 | 92.49 | 76.97 | 71.06 |
| ResNet34 (nasty) | 94.54 (-0.88) | 86.15 (-6.34) | 76.12 (-0.79) | 65.67 (-5.39) |

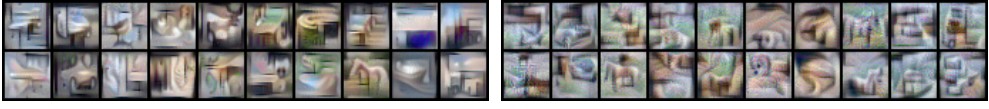

(a) Normal Teacher        (b) Nasty Teacher

Figure 6: Images generated by inverting a normal ResNet34 and a nasty ResNet34 trained on CIFAR-10 with DeepInversion. For each image, each column represents one category.

### 4.5 DISCUSSION

In practice, the owner of IP can release their sophisticated network defended by self-undermining training, at the cost of acceptable accuracy loss. As the aforementioned experiments show, even if a third-party company owns the same training data, they are not able to leverage knowledge distillation to clone the ability of the released model, since the performance of theirs would be heavily degraded, instead of being boosted as usual. Furthermore, we also show that stealers would suffer more than a $5\%$ drop of accuracy if data-free knowledge distillation is performed, of which performance drop is not acceptable in highly security-demanding applications, such as autonomous driving.

To sum up, our self-undermining training serves as a general approach to avoid unwanted cloning or illegal stealing, from both standard KD and data-free KD. We consider our work as a first step to open the door to preventing the machine learning IP leaking, and we believe that more future work needs to be done beyond this current work.

## 5 CONCLUSION

In this work, we propose the concept of *Nasty Teacher*: a specially trained teacher network that performs nearly the same as a normal one but significantly degrades the performance of student models that distill knowledge from it. Extensive experiments on several datasets quantitatively demonstrate that our nasty teachers are effective under either the setting of standard knowledge distillation or data-free one. We also present qualitative analyses by both visualizing the output of feature embedding and logits response. In the future, we will seek other possibilities to enlarge the current gap so that the proposed concept could be generally applied in practice, for which our current work just lays the first cornerstone.

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
