# OpenReview forum: "Undistillable: Making A Nasty Teacher That CANNOT teach students"
_ICLR.cc/2021/Conference — ICLR 2021 Spotlight_

### Official Review · AnonReviewer4 · 2020-10-19
**This is overall an ok submission.**

**Rating:** 7
**Confidence:** 4

**Review:**

This paper proposed a new method that prevent the teacher model from being learned by other models. The paper  introduces and investigates a concept called Nasty Teacher: a specially trained teacher network that yields nearly the same performance as a normal one, but would significantly degrade the performance of student models learned by imitating it.

Pros:

- This is overall a meaningful direction of knowledge distillation, that makes a neural network 'undistillable'.

- The method is simple yet effective by disturbing the output distribution of the teacher network.

- The experimental results show that the proposed method can succesfully prevent student networks from learning knowledge from teacher network both with original KD method and with data-free KD method.

Cons:

- Since this is a paper that focus on the real-world application problem, I would wonder whether the results are acceptable on ImageNet dataset. I notice that a 2% degrad appears on ResNet-50 on Tiny-ImageNet, which is almost unacceptable on real-world applications. I believe that there will be a larger gap between nasty teacher and original teacher on ImageNet in order to achieve the same undistillable results.

- The paper only discuss the influence of the final layer. What will happen if the student network learns from the feature maps deriving from the middle layers of teacher network? It seems that the loss function can only disturb the final output distribution, but we can still distill knowledge from the other layers. Experiments should be conducted for further explaination.

- It seems that the \alpha in Eq.(1) will also influence the student performance when learning from nasty teacher. The paper only shows that a large \alpha (=0.9 in the paper) will degrad the student performance. However, in practice the weight is always set to multiple values and the best among them is picked. I think the author should make an ablation study on \alpha.

---

> ### Author Response · Authors · 2020-11-20
> **Author Response to Reviewer #4 [Cons 1-3]**
>
> [Response to Cons 1:] In our experiments, we simply apply $\omega$ as 0.01 to all nasty teacher networks on TinyImageNet for a fair comparison. Here, we list the experiments of ResNet50 on TinyImageNet with smaller $\omega$ (0.0095), and the produced nasty ResNet50 only suffered less than 1% accuracy degradation.
>
>
> |  Network      | Teacher Performance | Student: ShuffleNetv2 | Student: MobileNetv2 |
> |:-:|:-:|:-:|:-:|
> | Student Baseline  | - | 55.74% | 51.72% |
> | ResNet50 (normal)  | 62.01%   |   58.01% (+2.27%)  | 54.18% (+2.46%) |
> | ResNet50 (nasty) |    61.21%(-0.80%) |    52.58% (-3.16%) | 45.78% (-5.94%) |
>
>
>
> As for ImageNet experiments, it has been well-proven in [1] that most KD approaches become invalid in this dataset. Since we don’t target our work at solving normal KD problems on ImageNet, we don’t include its experiments at the current stage. However, we are pleased to fall back to ImageNet if an effective KD approach is proposed in this dataset.
>
> [Response to Cons 2:]  We acknowledge that avoiding model stealing from feature maps in the middle layer is an interesting topic. In our setting, we consider the teacher as a black box (e.g., an executable or API), thus the feature map in the middle layers is unavailable to students. According to the analysis of tSNE, our method appears to mainly disturb the weights of the final layer, while keeping the learned feature representation nearly the same as the normal one, to the best balance between its own normal performance and the “undistillability”.
>
> [Response to Cons 3:] We add additional ablation studies with respect to $\alpha$ as you suggested. Generally, a small $\alpha$ can help student networks perform relatively better when distilling from the nasty teacher. However, a small $\alpha$ also makes the student depend less on the teacher’s knowledge and therefore benefit less from KD itself. Therefore, the student cannot easily get rid of the nasty defense while still mincing effectively through KD, by simply tuning $\alpha$ smaller.
>
> [1] Cho, Jang Hyun, and Bharath Hariharan. "On the efficacy of knowledge distillation." Proceedings of the IEEE International Conference on Computer Vision. 2019.

---

> > ### Comment · AnonReviewer4 · 2020-11-23
> > **Thanks for the rebuttal**
> >
> > The authors address all of my concerns in the rebuttal, and I decide to keep my score unchanged.

---

### Official Review · AnonReviewer1 · 2020-10-23
**Novel research direction, interesting method, some counter model stealing experiments will be helpful**

**Rating:** 7
**Confidence:** 4

**Review:**

Summary:
This paper explores an interesting and novel research problem: how to make a teacher model undistillable. This can be a promising countermeasure to model extraction/stealing. The proposed Nasty Teacher approach is a two stage method, which first trains a good teacher network, then utilizes a self-undermining KD strategy to further distill the good teacher network to a bad one. Overall, this  Nasty Teacher approach can reduce the performance of the student network by ~5% in most cases. Although the performance decrease is not huge, the proposed approach is promising and can be a very useful baseline for this new research direction. A set of ablation and understanding experiments have also been conducted to support the effectiveness of  Nasty Teacher.

Strong points:
1. A new security scenario, and a novel defense approach against model stealing.
2. The proposed idea is well motivated and described.
3. The effectiveness of the proposed approach is substantiated by thorough experiments.

Weak points:
1. Shown in Fig.3 and Fig.4, the performance decrease of the student network is not very significant, and is slightly sensitive to the parameters. By finding the right parameter, the student may still achieve ok performance.  What would happen if the student uses temperature \tau < 4 in Fig.4 (a)?
2. In Eq. (1), isn’t the second P_T should be P_S? The cross entropy is defined on p?, but I thought p are the logits? Why no \sigma in the XE?
3. Eq. (1) indicates the distillation needs logits. It is ok to train the nasty teacher, but how about the student network? In the intro the authors argued that protecting “backbox” APIs is one of the main motivations of the proposed method, but this was not tested in the experiments. What would happen if the student is distilled with probabilities (not logits)? I believe the effectiveness will be even better if the student can only access probabilities.
4. It seems like the distillation experiments were all done on 100% training data? This setting is somewhat less suitable from model stealing since the attacker may not have 100% training data. What would happen if the attacker has only 10% of training data? In Table 5: Data-free KD, I think a performance decrease of ~6% may still be a successful stealing, considering the huge amount of money and effort it takes to train the teacher network.
6. How well is the proposed method compared to other defense methods against model stealing? Prior works in this field should also be reviewed, if there are any.

---

> ### Author Response · Authors · 2020-11-20
> **Author Response to Reviewer #1 [Weak Points 1-5]**
>
> [Response to weak point 1:] We conduct additional experiments as you suggested. The experimental results demonstrate that even if smaller temperatures ($\tau$ < 4) are picked, the performance of student networks learned from nasty teachers still consistently underperforms those learned from baselines. We will include these additional results into our final draft.
>
> [Response to weak point 2:] Thank you for pointing out the typo in our paper. We acknowledge that the $p_T$ should be $p_S$ in Eq.(1). P is the logit and a softmax function ($\sigma$) should be added in the $\mathcal{XE}$ term.
>
> [Response to weak point 3:] KD from probabilities is equivalent to KD from logits with temperature $\tau$ = 1. Thus we can get their performance from the ablation study on $\tau$ as you suggested. As the trend in the ablation study, we observe that the performance of student networks can also be degraded when they are learned from distilling probabilities.
>
> [Response to weak point 4:] As you suggested, we add experiments that vary the percentage of training examples from 10% to 90% and keep other hyper-parameters the same. The experimental results show that nasty teachers still consistently contribute negatively to student networks even if only 10% of training data are included.  We agree that a performance decrease of 6% is still a good stealing in some applications, while it may not be acceptable in other highly security-demanding applications, such as autonomous driving. We will include your thoughts and suggestions into our final draft, and seek other possibilities to enlarge the gap in future work.
>
> [Response to weak point 5:]  We appreciate your suggestion and we will collect a new paragraph: "Defense against model stealing " in related work. We also notice that none of the current defense approaches could be directly applied to KD-based model stealing, which is rather a practical threat.

---

> > ### Comment · AnonReviewer1 · 2020-11-23
> > **Thanks for the response**
> >
> > My concerns have been addressed during the rebuttal. My major concern was the model stealing setting. I believe the authors can easily include those experiments in the final version. I will increase my score and recommend acceptance of this paper. The "Nasty teacher" technique should be of broad interest to the deep learning community.

---

### Official Review · AnonReviewer3 · 2020-10-24
**Weak Accept**

**Rating:** 7
**Confidence:** 4

**Review:**

This paper’s main idea is refreshing and attractive: proposing a defensive method called nasty teacher, to avoid knowledge leaking or cloning through KD. A nasty teacher model is a specially trained network that yields nearly the same performance itself, while significantly degrading the performance of student models learned by imitating it. The standing point for machine learning IP protection is novel and hasn’t seen many discussions before. The introduction section motivated the study clearly and nicely. The method and experiments demonstrate a promising first step towards protecting machine learning IPs.

The writeup has a smooth logic but demands much more serious proofreading. Specifically, my initial expectation after reading the abstract and introduction parts was quite high (those seem to have been well revised and mature). However, starting from Section 2 the readability drops to a rather poor level. There are many typos that hamper the paper:
-	Sec 1, “maximizing difference between nasty teacher” – missing “the”. Section 3.1, “a αclose to 1” – “a: should be “an”. The authors need carefully check their usage of the articles in general: “a” versus “an”, missing “a/an/the”, or largely confusing them.
-	Sec 2, “Teacher-free KD framework (Yun et al., 2020) train - “train” should be “trains”. There are a few more missing “s”/”es”
-	Section 3.1, “The higher α is” – “higher” should be “larger”. The authors seem to have confused “high””large””big” and abused them randomly throughout the paper, etc.
-	Section 3.2, “we hope to build a kind of logit whose argmax is still the correct answers” – this sentence is grammatically wrong from end to end …
-	“the distribution of incorrect answers varies as much as possible to the real distribution over the dataset” – I guess that is in contrary to what you wanted to say. “varies as much as…” means “correlates well with”, but your method should expect them to differ from each other.
… and I could have placed a longer list. I urge the authors to maintain the same high standard of revision beyond the point of Section 1.

Another issue is while the authors stressed their training algorithm called self-undermining KD in the abstract and introduction, that algorithm is only very briefly introduced in the last paragraph of section 3.2 -- which I nearly missed. I strongly suggest the authors to use an algorithm table to make your main algorithm clearer and more visible.

Besides, the text in Section 3.2 also needs to be considerably re-organized and revised; currently it is hard to figure out the right focus and lacks clarity. For example, I cannot see whether the adversarial network is also updated during training.

From the draft, it is also unclear whether the authors will release their codes and models for reproducibility

Overall, this paper is technically novel and interesting, and can potentially generate a positive impact. But the draft quality is currently unsatisfactory. Careful proofreading and thorough revision are requested from the authors; otherwise my final recommendation cannot be positive.

---

> ### Author Response · Authors · 2020-11-20
> **Author Response to Reviewer #3**
>
> We really appreciate your suggestions and we will revise the manuscript in an effort to improve its clarity and reader friendliness as follows:
>
> - Check the usage of definite/indefinite articles, tense of verbs, and other grammar pitfalls
> - Perform careful proof-reading to avoid any possible confusions
> - Re-organized Section 3, to elaborate and concentrate on the self-attack algorithm
>
> As for the update rule during adversarial training, we make the parameters of the adversarial network fixed and only update the parameters of the nasty network. More details will be included in the revised manuscript.
> Our code base including a pre-trained nasty ResNet18 on CIFAR10 will be uploaded as supplementary materials in the revised version and we firmly promise to release all codes and pre-trained models upon paper acceptance.

---

> ### Comment · AnonReviewer3 · 2020-11-23
> **Raise score to accept**
>
> The authors address all of my concerns in the rebuttal, and I decide to raise my score to accept.

---

### Official Review · AnonReviewer2 · 2020-10-27
**Interesting idea**

**Rating:** 7
**Confidence:** 4

**Review:**

Summary:
This paper reveals and studies a new problem, that KD is posing a potential risk of intruding the intellectual property (IP) of released ML models or their training data. Even trained ML models are released only in “black boxes” (e.g., as executable or APIs, no open-sourcing codes), their functionalities can be largely replicated by KD through imitating input-output behaviors. I find this problem and idea very interesting, and of both social and legal importance.

Pros:
- The self-undermining training method seems to be a variant of adversarial training. It maximizes the K-L divergence between the nasty teacher logits and those produced by another adversarial network. The authors used a similar idea to born-again NNs and self-training KD, to generate adversarial attacks from another pretrained network of the same architecture; thus no extra model is necessary. The training method is conceptually simple but effective.
- The authors conducted many experiments on several datasets to demonstrate that the proposed method is effective on both standard KD and data-free KD. Especially, they show the KD-immunity w.r.t various student models from small to large.

Cons:
- Even the authors presented an ablation study, it remains unclear to me why this self-attack is a good choice? Is convenience the only reason, and might there be more benefits (e.g., are the attacks generated by the same architecture more “specialized”)?
- “While our nasty teacher targets to degrade the performance of all kinds of students networks through KD”, I would suggest the authors to seriously tone down claims like this (or present substantially more experiments)
- We need more understanding why the nasty teacher works. Looking at Figure 2, I cannot get any useful information why the nasty features are more detrimental to distillation, and why this is not reflected on t-SNE?

---

> ### Author Response · Authors · 2020-11-20
> **Author Response to Reviewer #2 [Cons 1-3]**
>
> [Response for Cons 1:] We naturally choose the same network architecture as its adversarial learning counterpart since we don’t want to make any assumptions on network architectures in our main experiment. As in Table 4, we vary the architectures of adversarial networks and observe that weak adversarial networks (e.g., Plain CNNs) lead to less effective nasty teachers. Meanwhile, although stronger networks contribute to more effective nasty teachers, the trade-off accuracy is saturated quickly and converges to self-attack ones. Without loss of difficulty, we consider “self-attack” adversarial training as a convenient fallback choice.
>
> [Response for Cons 2:] Thank you for pointing out the issue of our tones. We intend to say our method is effective for several teacher-student pairs. We will tone down similar claims in the revised version.
>
> [Response for Cons 3:] In the manuscript, we present the visualizations of learned features and logit outputs in Figure 1&2 to better understand nasty teachers. Firstly, we observe that the feature-space inter-class distance of nasty teachers and the normal ones look similar, and this aligns with our goal that nasty teachers should perform similarly to normal ones. Secondly, we notice that the logit output of nasty teachers usually consists of multiple peaks, where normal teachers consistently output a single peak. It terms out that the output of multi-peak logits misleads the learning from the knowledge distillation and degrades the performance of students. The visualizations summarize the struggling behavior of balancing between its own performance and the ‘undistillability’.

---

> > ### Comment · AnonReviewer2 · 2020-11-21
> > **Thanks for the rebuttal**
> >
> > Thanks for the authors' feedback. The new visualizations are helpful. It seems that I do not have extra technical concerns. I raised my rate.

---

### Author Response · Authors · 2020-11-24
**Updates for the rebuttal revision**

We sincerely thank all reviewers for unanimously recognizing our work's significance and merits. We have responded to all concerns below.  Here we give an overview of changes made in the revised version during the rebuttal period.

- Additional experiments: as AnonReviewer1 and AnonReviewer4 suggested, we added additional ablation studies with respect to the balance factor $\alpha$, the amount of training data, and small student temperature values like $\tau<4$.
- Additional related work:  Following the suggestion from AnonReviewer1, we also collected a paragraph in the related work to discuss current defense methods against model stealing.
- Additional tSNE: To relieve the concerns from AnonReviewer2, besides the tSNE of feature embedding in our original draft, we also added the tSNE of logits to better understand nasty teachers.
- Reorganization of Section 3: Following the feedback from AnonReviewer3, we reorganized Section 3.2 and changed the notation to avoid any possible confusion.
- Improvement on writing quality: Following the feedback from AnonReviewer3, AnonReviewer2 and AnonReviewer4, we carefully check several grammatical issues and perform careful proofreading in the effort to improve the readability of the manuscript,
- Code: We upload our code base including a pre-trained nasty ResNet18 on CIFAR10 into the supplementary materials for reproductivity.

We thank all reviewers’ time again.

---

### Decision · Program_Chairs · 2021-01-07
**Final Decision**

**Decision:**

Accept (Spotlight)

**Comment:**

The paper argued some viewpoint about knowledge distillation quite interesting to me: the technically good KD might surprisingly be socially bad in helping outsiders "stealing" commercial models, even if the models are released as black boxes. Then the paper proposed a way called self-undermining KD in order to turn a well trained model into a "nasty teacher" (i.e., an undistillable model), and by this way the commercial models and the corresponding intellectual properties for training them from insiders can be nicely protected.

Overall, the quality is quite high. The argument is very conceptually novel and the method is still technically novel. The idea of the method is simple but works for the purpose --- that's great! Although the experimental significance seems not too impressive, the paper opens a door to a new world concerning model privacy instead of data privacy, and hence it is of social significance. In my opinion, the paper should have a potentially huge social impact to DL practitioners (and company owners), because KD is being used almost everywhere in the Internet industry to provide the standalone mode of Apps without clouds on personal devices. Based on the quality and the impact, I recommend to accept the paper as a spotlight presentation.